# Fumonisin B1 as a Tool to Explore Sphingolipid Roles in Arabidopsis Primary Root Development

**DOI:** 10.3390/ijms232112925

**Published:** 2022-10-26

**Authors:** Yanxue Zhao, Zhongjie Liu, Lei Wang, Hao Liu

**Affiliations:** 1State Key Laboratory of Crop Stress Adaptation and Improvement, School of Life Sciences, Henan University, Kaifeng 475000, China; 2Agricultural Genomics Institute at Shenzhen, Chinese Academy of Agricultural Sciences, Shenzhen 518000, China

**Keywords:** Fumonisin B1, sphingolipids, root development, stem cell niche, cell death

## Abstract

Fumonisin B1 is a mycotoxin that is structurally analogous to sphinganine and sphingosine and inhibits the biosynthesis of complex sphingolipids by repressing ceramide synthase. Based on the connection between FB1 and sphingolipid metabolism, FB1 has been widely used as a tool to explore the multiple functions of sphingolipids in mammalian and plant cells. The aim of this work was to determine the effect of sphingolipids on primary root development by exposing Arabidopsis (*Arabidopsis thaliana*) seedlings to FB1. We show that FB1 decreases the expression levels of several *PIN-FORMED* (*PIN*) genes and the key stem cell niche (SCN)-defining transcription factor genes *WUSCHEL-LIKE HOMEOBOX5* (*WOX5*) and *PLETHORA*s (*PLT*s), resulting in the loss of quiescent center (QC) identity and SCN maintenance, as well as stunted root growth. In addition, FB1 induces cell death at the root apical meristem in a non-cell-type-specific manner. We propose that sphingolipids play a key role in primary root growth through the maintenance of the root SCN and the amelioration of cell death in Arabidopsis.

## 1. Introduction

Plant root growth and development depend on the continuous production of cells in the root apical meristem (RAM), which harbors a stem cell niche (SCN) that supports continuous postembryonic organogenesis. In Arabidopsis (*Arabidopsis thaliana*), several key factors have been identified that are involved in the specification of the SCN and the transition from cellular proliferation to differentiation, including the two auxin-inducible AP2-type transcription factors *PLETHORA1* (*PLT1*) and *PLT2*, as well as the GRAS transcription factors *SHORT-ROOT* (*SCR*) and *SCARECROW* (*SHR*) [1,2,3]. PLTs function in a dose-dependent manner: high PLTs’ concentrations maintain the quiescent center (QC) and stem cell activity, intermediate concentrations regulate the division and differentiation of the transit-amplifying cells, and low concentrations allow cell elongation and cell differentiation [4]. The plant hormone auxin also plays major roles in the maintenance of cell division and the patterning of the root meristem [5,6,7]. Auxin efflux facilitators PIN-FORMED (PIN) family proteins generate and stabilize the auxin maximum and gradient, thus guiding root growth [8]. Being of a sessile nature, plants are continuously exposed to various environmental stress factors that can affect DNA replication or cause DNA damage [9,10,11,12]. Left unrepaired, DNA damage in actively dividing cells disrupts normal cellular functions and may thus severely affect plant growth and development [11,13,14,15].

Fumonisin B1 (FB1), an abundant mycotoxin, is the structural analog of the sphingolipids sphinganine and sphingosine and perturbs sphingolipid metabolism [16,17,18]. In mammals, FB1 inhibits the activity of all six known ceramide synthases, therefore causing a decrease in long-chain and very-long-chain fatty acid sphingolipid levels [19]. In Arabidopsis, FB1 selectively inhibits the ceramide synthases LAG ONE HOMOLOGUE1 (LOH1) and LOH3, resulting in a lower very-long-chain fatty acid ceramide and sphingolipids level [20,21]. FB1 has been widely used as a tool to explore the multiple functions of sphingolipids in mammals and plants [22,23,24,25,26].

Sphingolipids, enriched in the outer leaflet, comprise an estimated ~40% of the total lipids in the plant plasma membrane and are essential for eukaryotic life [21,27,28,29]. FB1-mediated sphingolipid metabolism remodels the membrane structure and influences numerous processes. Upon FB1 treatment, tobacco (*Nicotiana tabacum*) BY2 cells showed a severe drop in growth rate and delayed cell division, which was accompanied by the formation of endoplasmic reticulum (ER)-derived tubular aggregates as well as inhibited ER-to-Golgi cargo transport [30]. Sphingolipid metabolism has also been implicated in the mediation of programmed cell death in plants, as evidenced by sphingolipid biosynthesis mutants or FB1 treatments [25,31,32,33].

Recent studies have identified several mutants exhibiting a dwarf stature that has been attributed to altered sphingolipid profiles. The inhibition of lateral root emergence by reduced sphingolipid levels was correlated with the selective aggregation of the plasma membrane-localized auxin carriers AUXIN RESISTANT1 (AUX1) and PIN1 in the cytosol [20]. PASTICCINO2, a very-long-chain hydroxy fatty acyl-CoA dehydratase, interacted with cyclin-dependent kinase A, which is involved in the regulation of cell division [34]. The disruption of sphingolipid metabolism by FB1 treatment or mutants indicated that sphingolipids participate in root development [20,34,35,36]. The overexpression of the ceramide synthase gene *LOH2* also resulted in changed sphingolipid profiles and dwarf plants [37]. However, the precise mechanism by which sphingolipids regulate plant root development remains unclear.

Here, we used FB1 as a tool to explore the precise mechanism of action by which sphingolipids influence primary root development. We discovered that FB1 arrests primary root growth by reducing the expression of several auxin efflux facilitator *PIN* genes and the key stem cell niche-defining transcription factor genes *WOX5* and *PLT*s. Furthermore, FB1 induced cell death emergence in the RAM in a non-cell-type-specific manner.

## 2. Results

### 2.1. FB1 Inhibits Arabidopsis Primary Root Development

To assess the effect of FB1 in controlling primary root development, we monitored the growth rate of Arabidopsis primary roots after short-term FB1 exposure by transferring wild-type seedlings at four days after germination (4 DAG) to medium containing various FB1 concentrations, followed by incubation for three days. We observed that primary root elongation is strongly inhibited by FB1 concentrations of 1 µM and higher compared to untreated seedlings (Figure 1A,B). The root growth rate was comparable in seedlings treated with 1 or 2.5 µM FB1 during the first day, but the primary root nearly stopped elongating on the second day onward after being treated with 2.5 µM FB1. We also observed a more severe inhibitory effect on root elongation and an almost complete cessation of growth upon treatment with 5 μM FB1 (Figure 1A,B). Consistent with previous results [20], seedlings also showed a global inhibition of growth after transferring germinated seeds to FB1-containing medium for long-term exposure (Figure 1C,D). Thereafter, to minimize possible artifacts induced by prolonged FB1 exposure at the cellular level, we performed short-term treatments (16 h) with 2.5 μM FB1 for root development assays.

### 2.2. Transcriptome Analysis in Response to FB1 Treatment

For an in-depth investigation of the underlying molecular basis of FB1 effects, we performed transcriptome deep sequencing (RNA-seq) to identify differentially expressed genes (DEGs) between control and FB1-treated seedlings based on three biological replicates. We identified 5955 DEGs in response to FB1, with 3201 upregulated and 2754 downregulated genes (Figure 2A). A Gene Ontology (GO) term enrichment analysis of these DEGs, based on the biological process category, revealed a significant enrichment for the terms’ response to hypoxia, organonitrogen compound, root morphogenesis, oxidative stress, and phytohormone response, most of which are related to stress conditions (Figure 2B and Appendix A). Remarkably, the genes associated with root morphogenesis included the auxin efflux facilitator gene *PIN3*. These results were in line with phenotypic alterations of primary root growth in Figure 1 and previously reported data in response to FB1 [20].

### 2.3. FB1 Alters PIN and PLT Expression Levels in the Roots

In Arabidopsis root development, auxin forms a local gradient at the root tip that is generated by the auxin efflux facilitators PINs and regulates pattern formation and the orientation and extent of cell division [8,38,39]. The identification of DEGs enriched for the GO category of root morphogenesis prompted us to investigate whether the reduced primary root length was related to auxin efflux transporter genes. To this end, we determined the localization and expression of PINs upon FB1 treatment. As shown in Figure 3, green fluorescent protein (GFP) fluorescence derived from the reporter constructs *PIN1pro:PIN1-GFP*, *PIN2pro:PIN2-GFP*, *PIN3pro:PIN3-GFP*, and *PIN4pro:PIN4-GFP* was lower after treatment with 2.5 μM FB1 for 16 h.

In Arabidopsis, two main pathways regulate the activity of the RAM: the auxin-inducible *PLT1/PLT2* pathway and the *SHR/SCR* pathway [1,4,40,41]. PLT1 and PLT2, whose expression is strongly correlated with auxin gradients in the RAM, provide longitudinal information [4]. The *SHR/SCR* pathway provides positional information along the radial axis [40,41]. As shown in Figure 3, the accumulation pattern of PLT1 and PLT2, measured by the fluorescence of *PLT1pro:PLT1-YFP* and *PLT2pro:PLT2-YFP* reporter constructs, decreased by 25–40% upon FB1 treatment. By contrast, the abundance and localization of SHR and SCR were not affected by FB1, as indicated by the *SCRpro:H2B-YFP* and *SHRpro:SHR-GFP* reporter lines.

The RETINOBLASTOMA-RELATED (RBR) protein has been found to define the position of the asymmetric cell divisions in the stem cell area of the root through the modulation of the cell cycle regulator E2Fa [42]. We observed that the abundance of RBR transcript, together with E2Fa, was highly induced by FB1 (Appendix A). WEE1, one key cell cycle regulatory kinase that controls plant growth by arresting dividing cells in the G2-phase of the cell cycle in response to stress [43], was slightly induced by FB1 (Appendix A). Next, we investigated the cell cycle progression by the expression of cell cycle-related proteins. Among these tested genes, the expression of the Histone H4 gene, which is usually used as a marker of S phase cells, was obviously decreased upon FB1 treatment (Appendix A).

### 2.4. FB1 Induced Cell Death in the Root Meristem

Interestingly, we detected evidence of cell death for several root meristem cells upon FB1 treatment, as evidenced by propidium iodide staining (PI), which stains the cell wall and dead cells (Figure 4). To investigate the occurrence of cell death in greater detail, we stained the roots of seedlings with PI following treatment with FB1 for different durations and at various concentrations. We detected no PI-stained cells in the root meristem after growth in the presence of 1 μM FB1 for three days (Appendix A). Dead cells appeared in the RAM of 20% of seedlings after growth with 2.5 μM FB1 for one day; the number of dead cells increased with a longer incubation (Figure 4A and Appendix A). In addition, higher doses of FB1 produced more dead cells covering a larger root area in the meristem (Figure 4). Cells from the QC remained alive, but the expression of the QC marker *WOX5pro:ERGFP* (encoding ER-localized GFP) was strongly inhibited in the presence of FB1 (Figure 4D). The number of the primary root meristem was significantly smaller upon 1 μM FB1 treatment for three days (Figure 4E). After a longer FB1 exposure, we observed no death cells in the RAM and the QC cells divided (Figure 5).

The cell death phenotype in the meristem induced by FB1 may be corrected with the constitutive activation of DNA damage responses. To test this hypothesis, we measured the expression of the DNA damage response genes *ETHYLENE RESPONSE FACTOR115* (*ERF115*), *RADIATION SENSITIVE51* (*RAD51*), *BREAST CANCER SUSCEPTIBILITY1* (*BRCA1*), and *POLY(ADP-RIBOSE) POLYMERASE1* (*PARP1*) by reverse transcription quantitative PCR (RT-qPCR). ERF115 activity is needed when stem cells are damaged [44]. *RAD51* and *BRCA1* encode two DNA repair regulators that are involved in the repair of double-stranded DNA breaks, and the expression of *PARP2* is induced by ionizing radiation and radiomimetic drugs [45,46,47]. As shown in Figure 4F, the expression levels of all four genes were elevated after exposure to FB1.

## 3. Discussion

Sphingolipids not only function as structural components of membranes but also act as bioactive molecules involved in signal transduction and cell regulation in all eukaryotic cells [21,27,29]. However, little is known about the function of sphingolipids during plant root development. FB1 is a mycotoxin from *Fusarium moniliforme* that inhibits ceramide synthases and disturbs sphingolipid metabolism [20,22,48,49]. FB1 has been widely used as a tool to explore the multiple functions of sphingolipids in mammals and plants [22,23,24,25,26]. Sphingolipid metabolism disturbed by the application of 2.5 μM FB1 for 16 h was identical to that induced by a 9-day treatment with 0.5 μM FB1 [20]. Inspired by these studies, FB1 was chosen as a tool to explore the precise mechanism of action of sphingolipids in primary root development. The results obtained in this study indicated that FB1 repressed Arabidopsis primary root growth in a dose-dependent manner (Figure 1).

In Arabidopsis, auxin gradients are central to the identity of the QC and the SCN in the root meristem. Treatment with FB1 decreased the expression of the auxin output reporter *DR5:GUS* (where the β-*GLUCURONIDASE* [*GUS*] reporter gene is driven by the synthetic *DR5* promoter), mainly resulting from impaired PIN auxin transporters (Figure 3) [20]. In Arabidopsis, the auxin-inducible *PLT1/PLT2* pathway and the *SHR/SCR* pathway are the two main pathways responsible for root SCN maintenance [4]. The reduced expression of *PLT1* and *PLT2* by FB1 demonstrated that disturbed sphingolipid homeostasis may regulate the expression of the SCN-defining transcription factor *PLT* genes (Figure 3). *WOX5* is an important regulator that is specifically expressed in root QC cells to regulate the activity of distal SCNs [50]. Upon short-term treatment with FB1, *WOX5* expression was strongly repressed (Figure 4D). Under normal conditions, a low QC proliferation rate maintains the root structure and meristem function [10,44,50]. Longer FB1 treatment induced cell division in the QC (Figure 5). The RBR protein, which defines the position of the asymmetric cell divisions in the stem cell area of the root through the modulation of the cell cycle regulator E2Fa [42], was much higher by FB1 (Appendix A). WEE1, which controls plant growth by arresting dividing cells in the G2-phase of the cell cycle in response to stress [43], was also slightly induced by FB1. The expression of the Histone H4 gene, used as a marker of S phase cells, was obviously decreased upon FB1 treatment (Appendix A), indicating that the cell cycle progression was disturbed. From the data above, we conclude that sphingolipids may play an important role in maintaining the identity of the QC and stem cell activity.

As sessile organisms that are unable to escape from environmental hazards, plants utilize multiple pathways to cope with environmental stresses that can negatively affect DNA replication or cause DNA damage [10,11,51,52,53,54]. With its particular sensitivity to DNA damage induced by environmental stresses, the SCN has been shown to undergo cell death, thus protecting the genomic integrity and RAM activity to survive from severe stress [9]. Cell death triggered by γ irradiation, X-rays, or radiomimetic drugs such as bleomycin and zeocin in the root stem initial cells is cell-type-specific, with the stele stem cells (SSCs) being especially prone to entering the cell death program [9,44,55]. In most cases, chilling stress induces cell death in columella stem cell daughters [56]. Actinomycin D-induced cell death prefers the SSCs and stele cells in the RAM [57]. As opposed to the cell-type-specific cell death described above, we observed that FB1 induced cell death in the RAM in a non-cell-type-specific manner. Higher doses of FB1 caused more dead cells in the meristem (Figure 4). Regeneration programs are activated by DNA damage-induced cell death, after which QC cells divide in response to the activation of ERF115 [44]. The divided QC cells and induced expression of *ERF115* by FB1 suggest that the ERF115-dependent activation of QC cell division may take place in response to FB1. Therefore, further experiments will be necessary to elucidate the precise mechanism of FB1-induced cell death in the RAM.

## 4. Materials and Methods

### 4.1. Plant Materials and Growth Conditions

The wild-type was represented by ecotype Columbia-0 (Col-0), and the transgenic lines were as follows: *PIN1pro:PIN1-GFP* [58], *PIN2pro:PIN2-GFP* [59], *PIN3pro:PIN3-GFP* [60], and *PIN4pro:PIN4-GFP* [8]; *SCRpro:H2B-YFP* [61]; *SHRpro:SHR-GFP* [62]; *PLT1pro:PLT1-GFP* and *PLT2pro:PLT2-GFP* [63]; *WOX5pro:ERGFP* [8].

Seeds from the Arabidopsis (*Arabidopsis thaliana*) Columbia-0 (Col-0) accession were used as experimental materials. Seeds were surface-sterilized for 10 min with NaClO, washed five times with sterilized water, plated on half-strength Murashige and Skoog (MS) medium (containing 1% [*w/v*] sucrose and 1% [*w/v*] agar, pH 5.8) in the dark for 2–4 d, and then transferred to a standard plant incubator at 22 °C with a 16-h-light/8-h-dark photoperiod [64].

### 4.2. Chemical Treatments and Root Growth Analysis

For chemical treatments, 4-DAG (days after germination) seedlings were moved to fresh half-strength MS plates containing FB1 at the indicated concentrations and placed vertically for various time points for short-time treatment, or seeds were germinated on the half-strength MS plates containing FB1 at the indicated concentrations for long-term treatment. The same stock solution of FB1 was used throughout. For the quantification of root growth, seedlings were scanned with a root scanner, and then the root length was measured using ImageJ 1.52p (http://imageJ.nih.gov/ij).

### 4.3. Confocal Microscopy

Propidium iodide (PI) staining was performed as previously described [10]. For confocal microscopy, fluorescence in roots was detected using a Zeiss LSM980 laser scanning microscope (ZEISS, Oberkochen, Germany). The PI signal was visualized using 561 nm as the excitation wavelength and 591 to 635 nm as the emission wavelengths. GFP fluorescence was detected using 488 nm as the excitation wavelength and 510 to 530 nm as the emission wavelengths. YFP fluorescence was detected using 514 nm as the excitation wavelength and 530 to 600 nm as the emission wavelengths. Images and fluorescence intensities were processed using Zen 2012.

### 4.4. Reverse Transcription Quantitative PCR Analyses

For the reverse transcription quantitative PCR (RT-qPCR) analysis, total RNA was extracted from the roots of 7-day-old seedlings treated without or with 2.5 μM FB1 for 16 h using the RNeasy plant mini kit (QIAGEN, Hilden, Germany). RNA was treated with DNaseI and reverse-transcribed using a PrimeScript RT reagent kit (Takara, Kusatsu, Japan). After the RT reaction, the complementary DNA (cDNA) template was subjected to PCR in a 20-µL reaction using the SYBR Premix (Vazyme Biotech Co., Ltd., Nanjing, China) on an Applied Biosystems 7500 Real-Time PCR System. Three replicates were performed. Three reference genes, including *ACTIN2* (At3g18780), *ACTIN7*(At5g09810), and *UBQ5* (At3g62250), were analyzed to determine the suitable reference gene. The overall stability of the tested reference genes was measured by calculating the gene expression stability (M-value). *ACTIN7* had the best M-value and was therefore selected as a reference gene for normalization. Data presented are averages from three biological replicates ± standard deviation (SD). The primer sets are listed in Appendix A.

## 5. Conclusions

Few studies have focused on the role of sphingolipids in plant root development. Our findings enrich the understanding of sphingolipid functions during root development. We demonstrated that FB1, a useful tool for exploring the multiple functions of sphingolipids, affected primary root growth by reducing the expression levels and patterns of SCN-defining transcription factor genes. In addition, FB1 induced cell death in the RAM in a non-cell-type-specific manner. Given the interconnections between auxin, PLTs, PINs, the ERF115-mediated cascade, and the DNA damage response, it is noteworthy to identify key components of the root SCN response to stress. Therefore, it would be interesting to identify genes involved in sphingolipid metabolism with functions in root development, especially via the regulation of stem cell niche-defining transcription factor genes and DNA damage response genes.

## Figures and Tables

**Figure 1 ijms-23-12925-f001:**
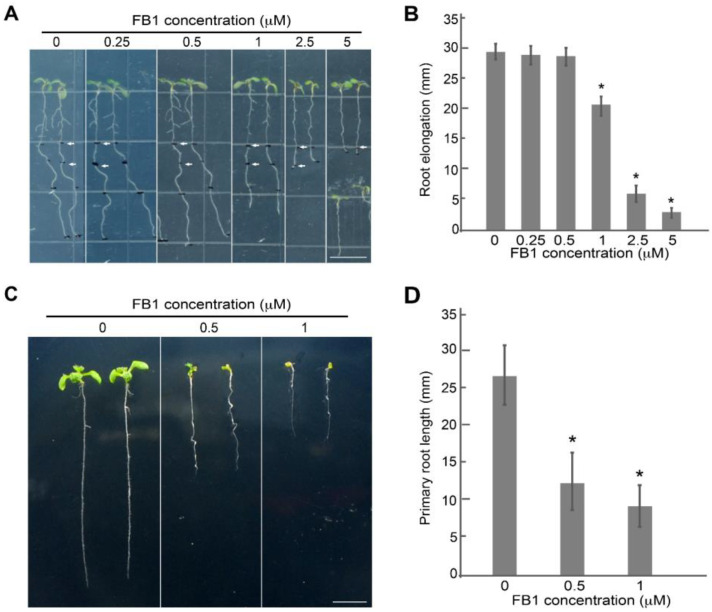
FB1 represses primary root growth. (**A**) Phenotypes of 4-DAG wild-type seedlings transferred onto half-strength MS medium (0) or half-strength MS medium containing 0.25, 0.5, 1, 2.5, or 5 μM FB1 for 3 days. The white arrowhead indicates the growth length of seedlings after transfer to FB1 medium for one day. Scale bar, 1 cm. (**B**) Primary root growth of wild-type seedlings after transfer to FB1 for 3 days in (**A**). Data are means ± standard deviation (SD) (*n* = 15–20). (**C**) Representative images of Arabidopsis seedlings grown in the presence of various concentrations (0, 0.5, or 1 μM) of FB1 for 7 days. Scale bar, 1 cm. (**D**) Primary root length of wild-type seedlings grown in FB1 for 7 days in (**C**). Data are means ± SD (*n* = 15–20). Asterisk (*) denotes significant difference relative to the seedlings without FB1, as determined by *t*-test; *p* < 0.01.

**Figure 2 ijms-23-12925-f002:**
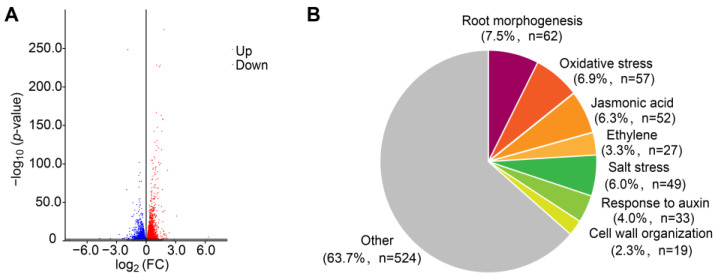
Whole-transcriptome analysis following FB1 exposure. (**A**) Volcano plot showing differentially expressed genes (DEGs) in wild-type seedlings treated with 2.5 μM FB1 compared to control seedlings. The horizontal line indicates the significance threshold for DEGs (*p* < 0.05). Upregulated and downregulated genes are shown with blue and red dots, respectively. (**B**) Enriched GO terms among DEGs. GO functional analysis of DEGs upon treatment with 2.5 μM FB1 with a |Log_2_FC| > 0.585 (i.e., 935 genes). ShinyGo v0.61 software was used for the GO enrichment analysis.

**Figure 3 ijms-23-12925-f003:**
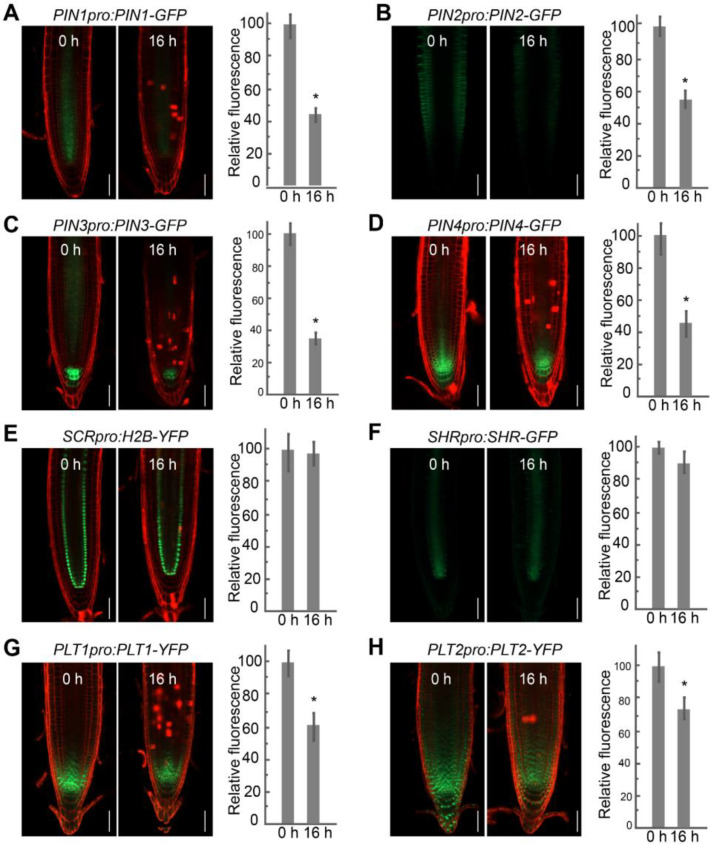
FB1 affects the abundance of PINs and PLTs. The localization pattern and quantification of fluorescence intensity (green) of PINs and stem cell niche markers treated with 2.5 μM FB1 for 0 and 16 h, respectively. The root tips were stained with propidium iodide (PI) (red). (**A**) *PIN1pro:PIN1-GFP*. (**B**) *PIN2pro:PIN2-GFP*. (**C**) *PIN3pro:PIN3-GFP*. (**D**) *PIN4pro:PIN4-GFP*. (**E**) *SCRpro:H2B-YFP*. (**F**) *SHRpro:SHR-GFP*. (**G**) *PLT1pro:PLT1-GFP*. (**H**) *PLT2pro:PLT2-GFP*. Data are means ± SD. Asterisk (*) denotes significant difference relative to seedlings without FB1, as determined by *t*-test; *p* < 0.01. Scale bars, 50 μm.

**Figure 4 ijms-23-12925-f004:**
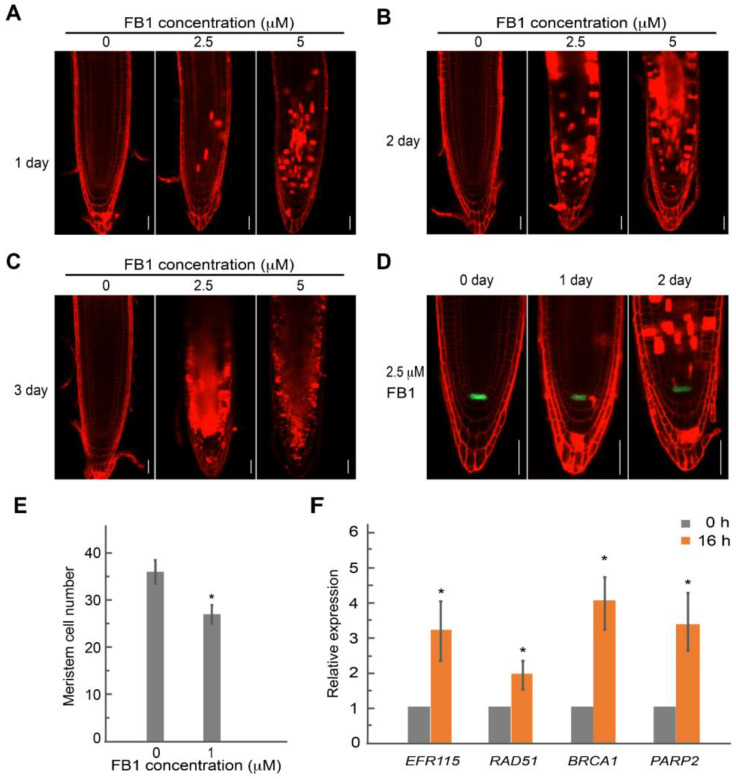
FB1 induces cell death in the root apical meristem. The root tips were stained with PI (red). (**A**) to (**C**) Confocal images of wild-type seedlings treated with FB1 at different concentrations (0, 2.5, or 5 μM) for (**A**) 1 day, (**B**) 2 days or (**C**) 3 days. (**D**) Fluorescence pattern of *WOX5pro:ERGFP* (green) in wild-type root tips treated with 2.5 μM FB1 for 0, 1, or 2 days. (**E**) Root meristem cell number of wild-type seedlings treated with 1 μM FB1 for 3 days. (**F**) Relative transcript levels of DNA damage response genes in wild-type seedlings treated with 2.5 μM FB1 for 0 or 16 h, as determined by RT-qPCR analysis. Data are means ± SD. Asterisk (*) denotes significant difference relative to seedlings without FB1, as determined by *t*-test; *p* < 0.01. Scale bars, 50 μm.

**Figure 5 ijms-23-12925-f005:**
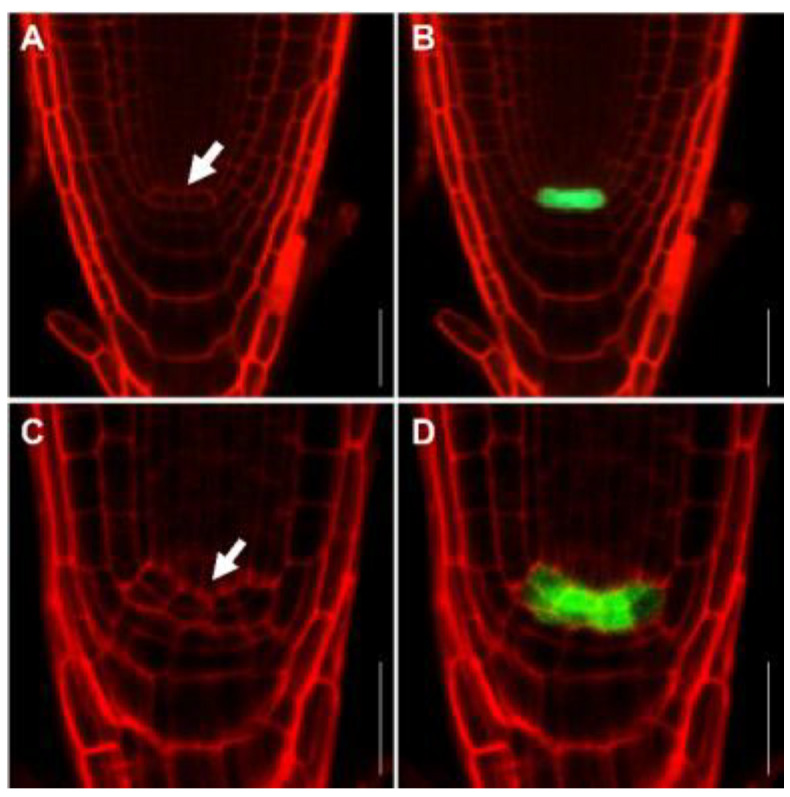
FB1 induces QC cells divided. (**A**,**B**) Expression pattern of *WOX5::ERGFP* (green) in wild-type seedling. (**C**,**D**) Expression pattern of *WOX5::ERGFP* (green) in wild-type seedling grown in half-strength MS containing 0.5 μM FB1 for 9 days. The root tips were stained with PI (red). The arrowhead in (**A**,**C**) indicates the QC cells. Bars = 25 µm.

## Data Availability

The original data to this present study are available from the corresponding authors.

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
