# Peer review of "Fumonisin B1 as a Tool to Explore Sphingolipid Roles in Arabidopsis Primary Root Development"

_ijms, 2022, doi:10.3390/ijms232112925_

Round 1
Reviewer 1 Report
The article by Zhao et al., if of interest for the effects of fumonisin1 on Arabidopsis primary root development. The authors tested hypothesis that sphingolipids regulate plant root growth by using ceramide synthase inhibitor FB1. However, I have a major concern that the article lacks the major information on sphingolipids itself.
The title of the article is “Fumonisin B1 as a tool to explore sphingolipid roles in Arabidopsis 2 primary root development”, but there is nowhere in the article that mentioned any quantification of sphingolipids in the root tips. The authors need to demonstrate that the biosynthesis of sphingolipids was indeed interrupted. Several methods such as MALDI should be used to visualize sphingolipids in those cells.
Figure 1: Please be specific on the number of replicates. For instance n > 10, please specify if it is n =10-20- or 10 -12, etc
Line 256: For more reliable qRT-PCR analysis, more than 1 reference genes should be used
Reviewer 2 Report
In this manuscript, the authors showed that FB1 could suppress the elongation of primary root of Arabidopsis and reduce the protein abundances of PINs and PLTs, and accelerate cell death. This provided evidences for further understanding the function of sphingolipids in root development. The manuscript is proper for publication in the ijms after revising the following concerns.
Major concern:
1. The main way for PINs to perform their function is to alter their subcellular location. The author should investigate the change of subcellular location of PINs after FB1 treatment.
2. The transgenic Arabidopsis plants such as PIN1pro:PIN1-GFP, PIN2pro:PIN2-GFP, PIN3pro:PIN3-GFP, PIN4pro:PIN4-GFP, SCRpro:H2B-YFP, SHRpro:SHR-GFP, PLT1pro:PLT1-GFP, and PLT2pro:PLT2-GFP are very important materials in the manuscript. However, there is not any introduction about these materials.
3. Based on the title, more key genes involved in root growth should be investigated in the article.
Minor concern:
1. Line 70, the aim of the first assay is not to assess the complex regulatory network controlling primary root development.
2. Line 75, the similar growth rate should be indicated in Figure1A.
3. From the data showed in the manuscript, it was not concluded that these results were in line with phenotypic alterations of primary root growth in Figure 1.
4. Line 124,The first abbreviation shall be indicated with full name, such as PLT1/PLT2.
5. What gene is H2B.
6. Line142, Figure3B should be Figure4B.
7. Line151,Figure4E showed the cell number, not the cell size.
8. Line189, the conclusion about “with the size of the primary root meristem being significantly reduced (Figure 1) is not correct.
9. Figure5,the magnification of Figure AB and Figure CD are different, but the ruler is same. Please check it.
Round 2
Reviewer 1 Report
The edits to the previous version of manuscript warrants publication in IJMS.
Author Response
Thank you for your approval.
Reviewer 2 Report
The manuscript has been improved. I am willing to agree that, with a minor revision, the manuscript is acceptable.
It has been reported that these genes such as PINs, H2B, SHR, PLT1, and PLT2 play important roles in root development. In this study, several transgenic Arabidopsis materials were used. However, there is not any introduction about these transgenic Arabidopsis plants in the materials and methods. They were provided by other laboratories or transformed by the authors. The native promoter was used to control the target gene in these plant expression vectors. If the length of this promoter is not clearly described, other researchers cannot repeat this experiment. If it comes from other scientists, you can cite their literature.
